# Fitness Trade-Offs between Phage and Antibiotic Sensitivity in Phage-Resistant Variants: Molecular Action and Insights into Clinical Applications for Phage Therapy

**DOI:** 10.3390/ijms242115628

**Published:** 2023-10-26

**Authors:** Jumpei Fujiki, Keisuke Nakamura, Tomohiro Nakamura, Hidetomo Iwano

**Affiliations:** 1Laboratory of Veterinary Biochemistry, School of Veterinary Medicine, Rakuno Gakuen University, Ebetsu 069-8501, Japan; j-fujiki@rakuno.ac.jp (J.F.); s21661158@g.rakuno.ac.jp (K.N.); t-nakamura@azabu-u.ac.jp (T.N.); 2Department of Medicine, University of California San Diego, La Jolla, CA 92093, USA; 3Phage Therapy Institute, Waseda University, Tokyo 169-8050, Japan; 4Research Center for Drug and Vaccine Development, National Institute of Infectious Diseases, Tokyo 208-0011, Japan; 5Department of Veterinary Medicine, Azabu University, Sagamihara 252-5201, Japan

**Keywords:** bacteriophage, antimicrobial resistance (AMR), fitness cost, phage cocktail, engineered phage, infectious disease, infection control, *P. aeruginosa*, *K. pneumoniae*, *A. baumannii*, *S. aureus*, evolution, arms race

## Abstract

In recent decades, phage therapy has been overshadowed by the widespread use of antibiotics in Western countries. However, it has been revitalized as a powerful approach due to the increasing prevalence of antimicrobial-resistant bacteria. Although bacterial resistance to phages has been reported in clinical cases, recent studies on the fitness trade-offs between phage and antibiotic resistance have revealed new avenues in the field of phage therapy. This strategy aims to restore the antibiotic susceptibility of antimicrobial-resistant bacteria, even if phage-resistant variants develop. Here, we summarize the basic virological properties of phages and their applications within the context of antimicrobial resistance. In addition, we review the occurrence of phage resistance in clinical cases, and examine fitness trade-offs between phage and antibiotic sensitivity, exploring the potential of an evolutionary fitness cost as a countermeasure against phage resistance in therapy. Finally, we discuss future strategies and directions for phage-based therapy from the aspect of fitness trade-offs. This approach is expected to provide robust options when combined with antibiotics in this era of phage ‘re’-discovery.

## 1. Introduction

Bacteriophages, also known simply as phages, are prokaryotic viruses that exclusively infect and kill bacteria. While phages were already being explored as antimicrobial agents in the early 1900s [1,2], the rise of antimicrobial chemical agents that became increasingly popular after the discovery of penicillin, overshadowed their use in Western countries [3]. However, after decades of being overlooked, the emergence of bacteria that are resistant to these antibacterial agents has refocused efforts on the use of phages for treating infectious diseases [4,5,6]. On the other hand, bacterial resistance to phages can also arise through a variety of molecular mechanisms. Several clinical studies on phage therapy have reported the occurrence of phage-resistant variants, which represents a significant concern for the successful development of phage-based therapies [7]. It is therefore important to carefully address phage resistance within the context of developing anti-bacterial treatments and therapies.

Phages have played a significant role in shaping the evolution of bacterial communities and populations through a co-evolutionary mechanism known as an arms race [8]. Recent studies investigating the fitness costs associated with phage resistance have highlighted how the selection of phage-resistant variants can have significant implications for clinical therapy settings [7,9,10]. This concept demonstrates that pathogenic bacteria can potentially develop resistance to phages, but that this occurs at the expense of virulence. Moreover, antimicrobial-resistant bacteria can regain susceptibility to antimicrobial agents as a fitness cost for phage resistance. These findings shed new light on the applications of phages to therapy and other phage-related approaches.

In this context, this review briefly outlines the basic virological properties of phages and their applications to date within the context of antimicrobial resistance. In addition, we summarize the occurrence of phage resistance in clinical cases. Notably, we review the fitness trade-offs between phage and antibiotic sensitivity, exploring the potential of evolutionary fitness cost as a countermeasure against phage resistance in phage therapy. Finally, we discuss future strategies and directions for phage-based therapy.

## 2. Virology of Phages

Phages are highly prevalent viruses that specifically infect bacterial cells. The global population of phages is estimated to be greater than 10^31^ particles, making them the most abundant biological entities on Earth [11]. Phages can be found wherever bacteria exist, including within the human body, and they play a crucial role in regulating bacterial populations as predators [12,13,14]. The ongoing evolutionary competition between bacteria and phages known as an arms race has driven the development of biological systems including microbial defense mechanisms against phages and anti-phage defense systems, contributing to the remarkable diversity observed in these microorganisms.

All phages package their genetic material within a capsid, and a significant number of phages possess a tail which is essential for attachment to host cells and for injection of their genome into the host [4,5,15]. Among the various types of phages, the most common are the tailed phages with double-stranded DNA (dsDNA) [4]. These can be classified into three distinct morphologies [15,16]: *Myoviridae* (contractile tail), *Siphoviridae* (long non-contractile tail), and *Podoviridae* (short tail). While the International Committee on Taxonomy of Viruses (ICTV) updated viral taxonomy in 2022 based on sequence information alone, eliminating morphology-based taxa, they acknowledged that these terms can still be used to describe phage characteristics and maintain their historical reference [17].

Phages primarily exhibit two main life cycle modes: the lysogenic and lytic life cycles [4]. Temperate phages employ both cycles, while virulent phages employ only the lytic cycle. During infection, phages attach to specific receptors on the bacterial surface and inject their genetic material into the bacterium. After gene expression and genome replication within the host, new phage particles are assembled and released by lysing the host cells through the action of endolysin, a peptidoglycan hydrolytic enzyme encoded in the phage genome [5,18]. While both temperate and virulent phages employ this lytic cycle, temperate phages can also enter a lysogenic state. In the lysogenic state, repressor proteins encoded by the temperate phages repress the genes responsible for bacterial lysis. Lysogenic phages integrate into the host chromosome or form independent replicating plasmids known as prophages. Repressors encoded in prophages prevent reinfection by the same phage. Under certain environmental conditions or stressors, the lysogenic phage genome can be excised, initiating the lytic cycle [19]. The characteristics of temperate phages, such as their capacity for horizontal transmission of genes associated with virulence and antimicrobial resistance among bacteria, as well as their tendency for maintaining a high frequency of lysogenic states, generally make them less suitable for therapeutic applications [4,19,20,21].

Bacterial host factors play a significant role in determining the specificity of phage infections. Bacteria have developed various anti-phage systems to counteract phage infection at various stages of the viral life cycle. For example, restriction-modification systems and CRISPR-Cas systems are well-known defense mechanisms that disrupt the phage genome before replication [22]. On the other hand, the phage host range is primarily restricted by receptors present on the surface of the host bacteria. Various structures on bacterial surfaces, such as outer membrane proteins, polysaccharides, teichoic acids, pili, flagella, and transporters, can act as bacterial receptors [23,24,25,26,27,28]. Much of the acquisition of phage resistance is attributable to mutations or deletions in receptor genes.

## 3. Phage Therapy within the Context of Antimicrobial Resistance

Although phages were first discovered by Frederick Twort in 1915 through experiments on filter-passing viruses, providing definitive evidence of their existence at the time was difficult [1]. Two years later, Felix d’Herelle isolated the first phage from the intestinal contents of dysentery patients, describing them as invisible living entities that parasitize bacteria [2]. D’Herelle recognized the therapeutic potential of phages and successfully used phages to treat patients with bacillary dysentery in 1921 [29,30]. Furthermore, a large clinical trial conducted in India by D’Herelle and colleagues in 1927 demonstrated a significant reduction in mortality rates among cholera patients treated with the *Vibrio cholerae* phage [31]. Based on these successes, scientists attempted numerous phage therapies for treatment of other bacterial infections. However, the outcomes of these phage therapy for other bacterial infections were inconsistent [32,33]. This could potentially be explained due to factors such as the lack of adequate controls, variations in phage dose and administration routes, the narrow host range of phages, bacterial-derived substances in phage products, and the influence of body fluids and the immune response [34]. Several of these are still discussed today [35,36]. Thereafter, in Western countries, the commercialization of antibiotics gradually replaced phage therapy. However, in countries such as the Soviet Union, Georgia, and Poland, where access to antibiotics was restricted by the “Iron Curtain” during the Cold War era, the use of phages to treat infectious diseases has continued [6]. Against this background, the emergence and increasing prevalence of antimicrobial-resistant bacteria has become a serious concern. It has been estimated that antimicrobial-resistant bacteria will be responsible for the deaths of approximately 10 million people annually in 2050 unless a global program to reduce antimicrobial resistance is implemented [37]. Consequently, phage therapy has been proposed as a potential “trump card” for treating these infections, given their unique bactericidal mechanisms that differ from those of antibiotics, making them effective for treating antimicrobial-resistant bacteria. In 2017, the Food and Drug Administration (FDA) approved the first application of phage therapy for treating infections caused by multidrug-resistant *Acinetobacter baumannii* [38]. In this clinical case, despite multiple antibiotics and percutaneous drainage of a pancreatic pseudocyst with suboptimal results, the use of phages significantly improved symptoms and ultimately led to a full recovery. After this, multiple case reports describing the application of phages to the treatment of multidrug-resistant bacterial infections have been published [4,39]. Recently, phage therapy programs have been established in the US, Belgium, France, and Sweden [4]. Notably, Belgium has initiated the implementation of a practical framework for phage therapy, focusing on creating personalized phage treatments using magistral preparations [40,41]. In addition, human clinical trials have been conducted for a variety of bacterial infections, as shown in Table 1. While one trial for treating otitis externa caused by *Pseudomonas aeruginosa* infection has shown positive therapeutic outcomes [42], other trials have failed to demonstrate conclusive efficacy [43,44]. In most clinical trials, the investigation and documentation of bacterial resistance to phage therapy have generally been inadequate, largely due to the lack of prior phage sensitivity tests. Taken together, phage therapy is a promising alternative approach to treating antimicrobial-resistant bacterial infections, and ongoing research and clinical trials are being conducted to further explore the efficacy of this treatment modality and establish its role in modern medicine.

## 4. Adverse Effects of Phage Resistance on Therapeutic Outcomes

Despite receiving considerable attention as a promising approach against antimicrobial-resistant bacterial infections, phage therapy faces challenges due to bacterial phage resistance; this is similar to the challenges facing antibiotics. Notably, several clinical studies on phage therapy have documented the occurrence of phage-resistant variants, and these findings have raised concerns about the successful development of phage therapies. Based on this perspective, we present a summary of phage therapy cases where phage-resistant variants were encountered, particularly instances that resulted in inadequate therapeutic outcomes or instances that necessitated changes in phage therapy protocols (Table 2).

In these clinical cases of phage therapy, each patient was treated with a combination of phages known as a “phage cocktail”. Generally, the purpose of using phage cocktails is to prevent the development of phage-resistant variants [4]. It has been observed that using a combination of phages that target different receptors can effectively suppress or delay the emergence of phage resistance [27,51,52]. In these cases, although evaluating the exact effectiveness of these phage cocktails in suppressing resistance was challenging, primarily due to the lack of clarity regarding the specific receptors targeted by each phage in the cocktail, phage resistance was still observed. After phage resistance was observed, in five out of eight cases at the University of California San Diego and Eliava Phage Therapy Center (Cases 1, 2, 3, 5, and 6; Table 2), the introduction of additional phages targeting the phage-resistant variants was required [38,39,45,46,48]. In two separate cases in China and Spain (Cases 4 and 8; Table 2), the emergence of phage-resistant variants mandated exploring alternative therapeutic strategies, which may or may not have incorporated phages [47,50]. In Case 7 shown in Table 2, Dedrick et al. demonstrated that, although phage resistance was not specifically concluded to be a major factor limiting the effectiveness of treatment, the development of a phage-resistant variant during treatment and the failure of intravenous mycobacteriophage therapy to cure refractory *Mycobacterium abscessus* lung disease was observed [49]. Taken together, these clinical outcomes strongly suggest that phage resistance had a significant negative impact on successful treatment. Notably, according to a report by Schooley et al. (Case 1), a phage-resistant variant was isolated from a patient just eight days after the initial administration of phage cocktails for *A. baumannii* infection [38]. Additionally, Bao et al. and Blasco et al. (Cases 4 and 8) described cases in which phage resistance developed within five days and one week from the beginning of phage administration against *Klebsiella pneumoniae* and *P. aeruginosa* infections, respectively [47,50]. These cases suggest that phage-resistant variants can emerge in a relatively short timeframe, even within approximately a week after administration in clinical phage therapy. However, as seen in the cases managed at the Eliava Phage Therapy Center (Cases 2 and 3), it appears that phage resistance can also develop gradually during relatively long-term phage administration spanning several months [45,46].

To address the challenge of phage-resistant variants, most cases required specific interventions, such as the preparation of alternative phage cocktails. In fact, personalized cocktails prepared during phage therapy are more likely to be effective, but a major concern is the time required to isolate (or screen), amplify, and purify new phages for phage resistance. In addition, difficulties in accessing sufficient phage banks or isolating novel phages against resistant variants would further complicate the production of personalized cocktails during phage therapy. Consequently, a proactive strategy that relies on a comprehensive molecular understanding of how phage-resistant variants develop could be a key for successful phage therapy.

## 5. Molecular Mechanisms of Fitness Trade-Offs between Phage and Antibiotic Resistance

Phage infection can induce surface mutations in pathogenic bacteria, creating conditions that lead to more positive treatment outcomes as a fitness trade-off for developing resistance. In particular, it is considered that phage resistance could potentially restore bacterial sensitivity to antibiotics [7,10], thereby extending the lifespan of existing antimicrobial agents and creating a strategy for treatment with antimicrobials with phages. Given that mutations in phage receptors contribute markedly to the fitness trade-offs between phages and antibiotic resistance, this section summarizes the molecular mechanisms of these trade-offs, focusing on two kinds of phage receptors: polysaccharides and drug efflux transporters.

### 5.1. Fitness Trade-Offs via Saccharide

Polysaccharides, the most ubiquitous polymers produced by bacteria, are important structural components of bacterial organisms. On the other hand, polysaccharides, such as exopolysaccharide antigens, capsular polysaccharides, and lipopolysaccharides (LPS), can also serve as major phage receptors [10,24,28]. Consequently, bacterial resistance to phages that specifically target saccharides may result in structural changes to these saccharides. Several studies have shown that modifications in these LPS could lead to increased susceptibility to antibiotics, particularly antimicrobial peptides.

Polymyxin B is a cationic antimicrobial peptide that binds to lipid A, a component of LPS that form the outer membrane of the cell wall. Once polymyxin B binds to lipid A, it alters the permeability of the cell membrane, disrupting both the outer and inner membranes, thus exerting antimicrobial activity [53]. It has been shown that susceptibility to polymyxin B is correlated with the degree of polymyxin B adsorption to LPS [54]. However, this susceptibility is inversely related to the presence of the cationic component Ara4N in LPS, which is attached to the phosphate groups in lipid A [55,56,57]. Furthermore, elevated expression of galactose as the terminal sugar in the primary oligosaccharide chain has been shown to contribute to polymyxin B resistance [57,58]. Therefore, it is expected that alterations to the core structure of LPS through phage resistance that is induced by LPS-targeting phages could affect polymyxin B susceptibility. Indeed, in *Yersinia pestis*, resistance to polymyxin B was observed to be 31–250-fold lower (MIC < 20 U mL^−1^) in strains with LPS synthase defects (e.g., *waaQ*, *waaE*, *waaF*, *waaC*, *hldE*, and *waaA* mutants), compared to wild-type strains. This decrease in resistance could be attributed to the limited presence of Ara4N within the LPS, which results in an incomplete inner core region [59]. Based on these molecular mechanisms, Filippov et al. reported that LPS core-targeting phages promoted the emergence of phage-resistant *Y. pestis* variants that had an altered LPS structure [60]. These phage-resistant LPS structures resulted in reduced virulence in mice due to shortened LPS lengths, and restored the susceptibility of *Y. pestis* to polymyxin B (Figure 1A).

*Enterococcus* phages are known to attach the *Enterococcus* polysaccharide antigen (Epa) as their primary receptor [61,62,63]. On the other hand, Dale et al. showed that susceptibility to daptomycin, an anionic lipopeptide, is enhanced by the deletion of the Epa synthesis gene, *epaO* [64]. Additionally, Ho et al. observed that mutations in *epaR* led to increased daptomycin susceptibility in *E. faecalis* [62]. While the precise molecular mechanisms underlying this shift in susceptibility are not fully understood, Canfield et al. proposed that the variability in susceptibility to daptomycin among *Enterococcus* Epa mutants may be due to changes in the presence of teichoic acids on the cell surface [65]. This hypothesis is supported by the recent discovery that the *epa* variable genes are involved in teichoic acid biosynthesis [66]. It is further supported by observations in *Staphylococcus aureus*, where disruptions in metabolism resulting in increased production of teichoic acid or d-alanylation of teichoic acid were reported to be correlated with daptomycin tolerance [67,68,69]. Similarly, inducing mutations in *lafB*, which encodes lipoteichoic acid glycosyltransferase, induced a daptomycin-hypersusceptible phenotype in *Enterococcus faecium* [70]. Moreover, mutation of *bgsB*, which functions alongside a *lafB* homolog (*bgsA*) in lipoteichoic acid anchor biosynthesis, was found to lead to enhanced susceptibility to daptomycin in *E. faecalis* [62]. Thus, one possible molecular mechanism may be that resistance to *Enterococcus* phages triggers mutations in the *epa* genes, which in turn results in the incomplete synthesis of lipoteichoic acid and restoration of daptomycin susceptibility (Figure 1B).

It has been reported that phage-resistant strains of *A. baumannii* lack capsular polysaccharide, which has been identified as the primary receptor for phages [38,71,72]. Notably, genes in the K locus of *A. baumannii* regulate the production, modification, and export of capsular polysaccharides. Altamirano et al. reported that infection of ΦFG02-R AB900 led to selected mutations in *gtr29* within the K locus, which encodes for a glycosyltransferase. Similarly, ΦCO01-R A9844 infection selected mutations in *gpi*, which encodes for the enzyme glucose-6-phosphate isomerase. These phage-resistant variants exhibited increased colistin sensitivity [72], likely due to the loss of the capsule, which facilitates the diffusion and eventual insertion of colistin into the membrane [73]. The absence of the capsule material might allow the colistin molecule to diffuse directly to the membrane without having to cross the capsule barrier (Figure 1C).

With respect to fitness trade-offs between phages and non-peptide antibiotics, interesting molecular mechanisms have been proposed that involve structural changes in the LPS of *Escherichia coli*. RfaP proteins catalyze the phosphorylation of heptose I in the bacterial outer membrane [74]. Therefore, when *rfaP* is absent or inhibited, the lack of phosphorylation of heptose I increases membrane permeability, allowing hydrophobic antibiotic agents to enter the cell. As a component of the bacterial outer membrane, the RfaP protein is also involved in the pathway for LPS core biosynthesis [75]. Consequently, mutations in *rfaP* can lead to resistance to LPS-targeting phages and increased sensitivity to hydrophobic antibiotics [76]. Through this mechanism, the restoration of susceptibility to hydrophobic antimicrobial agents, such as chloramphenicol, kanamycin, and novobiocin, has been observed in a phage-resistant, *rfaP* mutant strain of *E. coli* (Figure 1D).

### 5.2. Fitness Trade-Offs via Drug Efflux Transporters

In addition to the polysaccharides described in the previous section, membrane proteins can also act as phage receptors [10,22,28]. Among these, drug efflux pumps present a particularly attractive target for inducing fitness trade-offs because the molecules acting as phage receptors themselves contribute directly to antibiotic resistance. When the drug efflux pump is mutated or deleted due to phage resistance, it is expected to lead to a corresponding reduction in drug efflux capacity. This concept is based on a very well defined molecular mechanism that was first reported using the ΦOMKO1 against *P. aeruginosa* [77]. In 2017, Chan et al. screened 41 *Pseudomonas* phage isolates using knockout strains of *P. aeruginosa* PAO1 lacking the outer membrane porin M (OprM) to identify phages with significantly reduced infection titers. As implied by the results of this screening approach, ΦOMKO1 is likely to interact directly or indirectly with outer membrane OprM, a component of the MexAB and MexXY drug efflux systems. Moreover, they showed that ΦOMKO1-induced resistance in clinical isolates of *P. aeruginosa* increases susceptibility to ciprofloxacin, tetracycline, ceftazidime, and erythromycin by up to 50 fold in vitro, as well in infection experiments with the moth *Galleria mellonella* [77,78]. Hence, resistance induced by phages targeting drug efflux pumps, such as ΦOMKO1, would be expected to induce evolutionary changes that alter the function of the efflux pump through genetic mutations or deletions, ultimately leading to reduced antibiotic efflux from the bacteria (Figure 2A).

It has also been suggested that the drug efflux pump is an ideal target for *E. coli* [79]. The *Escherichia* virus ΦU136B, which recognizes the inner core of LPS and the drug efflux transporter TolC, was screened using *E. coli* strains with a *tolC* knockout, similar to the strategy employed by Chan et al. in the case of ΦOMKO1 [77]. Whole-genome sequencing of ΦU136B-resistant variants of the K-derived laboratory strain BW25113 revealed that 100% of these mutants displayed *tolC* mutations or mutations related to LPS synthesis. Moreover, all six *tolC* mutants exhibited lower resistance to tetracycline, which is a substrate for TolC, compared to the parental strain. Conversely, it was also demonstrated that certain phage-resistant strains possessed mutations only in LPS synthesis-related genes. In such cases, restoration of colistin sensitivity occurred while tetracycline resistance increased. The recovery of colistin sensitivity may be attributed to increased accumulation of the antibiotic in the membrane via structural changes in the LPS core. Additionally, a phage targeting TolC and the LPS inner core is known in *Salmonella enterica* serovar Enteritidis [27]. Using phage GSP032, Gao et al. demonstrated that ΦGSP032-resistant variants have similarly been associated with enhanced susceptibility to colistin, erythromycin, doxycycline, gentamicin, and ceftiofur. These effects likely arise from mutations that attenuate TolC function and disrupt the LPS core structure (Figure 2B).

### 5.3. Fitness Trade-Offs via Large Chromosomal Deletions

Bacteria also have DNA mismatch repair systems, and the enzyme Mut is known to play a central role in this mechanism. When Mut cleaves DNA at two or more sites, the fragments are deleted from the bacterial chromosome. It has been reported that *P. aeruginosa* acquires resistance to LPS-targeting phages through Mut-mediated chromosomal deletion when a gene essential for LPS core synthesis, *galU*, is included in the deleted region [80] (Figure 3). In addition, *hmgA*, which codes for an enzyme that metabolizes homogentisic acid located near *galU*, is also deleted at the same time during chromosome deletion. As a result, *P. aeruginosa* that cannot metabolize homogentisic acid exhibits brown pigmentation [80,81,82,83]. Shen et al. investigated phage resistance selection via MutL and referred to the resulting brown-pigmented phage-resistant variants as “brown mutants” (brmts) [80]. On the other hand, since the deleted regions are often large (i.e., up to several hundred kbp), it is assumed that the selection of phage-resistant variants by this mechanism will lead to a fitness trade-off in terms of antibiotic susceptibility if the large deleted chromosomal sequences encode genes related to antibiotic resistance. This idea was popularized after a report by Nakamura et al. who used LPS-targeting *P. aeruginosa* virus ΦS12-3 [84]. They focused on examining what other kinds of antibiotic-resistant genes exist in the large deleted chromosomal regions that confer phage sensitivity in a *galU*-dependent manner, and referred to these fluctuating sequences as the Bacteriophage-induced *galU* Deficiency (BigD) region. Eventually, they found that the genes encoding multidrug efflux pumps, *mexX* and *mexY*, are located between *galU* and *hmgA* in the BigD region of a phage-resistant variant of a *P. aeruginosa* veterinary isolate. Thus, phage-resistant variants that exhibit brown pigmentation are likely to lack MexXY, and they suggest that this feature is clinically useful as a marker as it can be used to effectively predict the likelihood of a fitness trade-off. They also reported increased sensitivity to enrofloxacin and orbifloxacin, which are known substrates of MexXY, in brmts obtained by co-culture with ΦS12-3. It has been demonstrated that even when the drug efflux pump is not the direct receptor of phages, it is possible to attenuate the function of the drug efflux pumps using an indirect molecular mechanism such as BigD (Figure 3). The results also suggest that more widely used LPS-targeting phages, compared to drug efflux pump-targeting phages, can also be used to indirectly induce a fitness trade-off via drug efflux pump function.

Phage-resistant variants with large chromosomal deletions can also be selected by drug efflux pump-targeting phages. Valappil et al. isolated phage PIAS with receptors for MexY and OprM and demonstrated that ΦPIAS infection selected phage-resistant variants with large chromosomal deletions, including *mexXY* in *P. aeruginosa* [85]. Other phage-resistant variants against ΦPIAS without large chromosomal deletions showed nonsynonymous substitutions in MexY. Based on these results, they also reported that the combination of ΦPIAS with the MexXY substrates fosfomycin, gentamicin, tetracycline, and ceftazidime had an improved inhibitory effect on the development of phage-resistant variants compared with phage only. In addition, Menon et al. also obtained brmt in PAO1 through co-culture with phages and examined its properties [86]. They reported that the brmt of PAO1 showed deletions of chromosomal regions that included *galU*, *mexXY*, and *hmgA*, which are hypersensitive to antimicrobial peptides LL-37 and colistin compared to the parental strain. These amphipathic antimicrobials can bind to LPS and destabilize negatively charged bacterial cell membranes in *P. aeruginosa*, resulting in membrane permeabilization and cell lysis. Given that increased adsorption of antimicrobial peptides on bacterial surfaces has been observed, it is suggested that an increase in the negative surface charges associated with genomic deletions likely contributes to the increased susceptibility to antimicrobial peptides.

In this manner, large-scale chromosomal deletions have been revealed as the molecular basis for fitness trade-offs between phages and antibiotic resistance, mainly in *P. aeruginosa*, but a similar approach may be possible for pathogens such as members of Enterococci, which can also acquire phage resistance via Mut [61,87].

## 6. Synergistic Interactions between Phages and Antibiotics in Phage Therapy

All of the clinical cases listed in Table 2 showed the development of phage-resistant variants. On the other hand, in two of these eight cases, antibiotics were also used after the emergence of phage-resistant variants and a therapeutic effect was observed [47,50]. In Case 4, although recurrence of phage resistance was observed in phage therapy targeting a urinary tract infection caused by *K. pneumoniae*, a successful outcome was achieved through a combination of phage and sulfamethoxazole-trimethoprim [47]. Interestingly, although *K. pneumoniae* strains were completely resistant to sulfamethoxazole-trimethoprim before phage therapy, the combined administration of sulfamethoxazole-trimethoprim and a phage cocktail prevented the emergence of phage resistance in vitro and resulted in favorable clinical outcomes. In Case 8, when the phage cocktail was administered along with ceftazidime-avibactam, phage-resistant variants developed and treatment was not successful [50]. Subsequently, ciprofloxacin, to which susceptibility was restored in phage-resistant variants, was administered in combination with amikacin and ceftazidime-avibactam, resulting in therapeutic efficacy in the patient. In addition to the cases shown in Table 2, there are clinical cases that demonstrate the synergistic effects of phages and antibiotics (Table 3). For example, in Cases 7 and 10 shown in Table 3, phage-resistant variants were observed in phage therapies; however, favorable clinical results were obtained without specific measures against these phage-resistant variants (e.g., development of new phage cocktails) [39,88,89]. These cases suggest that the favorable outcomes observed could be attributed to the combination treatment with antibiotics. The ΦOMKO1, which targets OprM (see Section 5.2), has also been shown to be effective in clinical trials as expected [90]. Unlike in other cases, such as 7, 10, and 11, where multiple antimicrobial agents were employed [39,88,89,91], ceftazidime alone was used in this case, yielding favorable results. This could be due to the molecular understanding of the fitness trade-off between ΦOMKO1 and specific antibiotic resistance in advance. In addition, it is noteworthy to mention the clinical reports from the Hannover Medical School, which showed considerable variability in terms of findings; up to 2023, 33 cases of personalized phage therapy were reported, of which 31 yielded successful clinical outcomes [92]. Of these, Rubalskii et al. reported the details of several cases related to cardiovascular surgery [93]; 7 cases describing the control of pathogenic bacteria are shown in Table 3. In one of these cases (Case 6), phage therapy was not successful. Since neither the development of phage-resistant variants nor the production of neutralizing antibodies against phages was observed, it is assumed that phage access to the site of infection was a major contributing factor. The high success rate of cases involving phage therapy at Hannover is considered to be largely due to the combination of phages and traditional antimicrobial agents [92].

## 7. Future Perspectives in Phage Therapy

The development of phage resistance is a significant concern in phage therapy and constitutes a hurdle that needs to be overcome [7]. However, as emphasized in the present review, it may be possible to induce the development of favorable phenotypes for therapy by directing pathogenic bacteria to evolve in a specific manner. Within the context of fitness trade-offs, affecting the susceptibility of bacteria to antibiotics using phages is a potentially promising strategy. To achieve clinically useful outcomes, it is therefore very important to clarify the mechanisms underlying these trade-offs at a molecular level. It is considered that such an understanding will facilitate the formulation of optimal phage-antibiotic combinations and guide the selection of suitable antibiotics in the presence of phage-resistant variants in the future. Further, in addition to the trade-off in antibiotic susceptibility, a trade-off in virulence due to phage resistance has been reported [7,9]. Consequently, synergistic effects arising from these influences may be anticipated.

The challenge associated with inducing fitness trade-offs lies in their generality and frequency of occurrence. Since case studies are still based primarily on limited combinations of specific phages and host bacteria, it is necessary to validate the fitness trade-off mechanisms using a larger set of clinical isolates and to validate trade-off strategies using bacterial strains from diverse backgrounds. In addition, as evidenced by studies on TolC-targeting phages, which also recognize LPS, phage-resistant variants are not always selected as the desired receptor mutation. Burmeister et al. reported that phage-resistant variants with mutations in only the LPS core, and not in TolC, did not show ideal recovery of antibiotic susceptibility [79]. Large-scale chromosomal deletions mediated by Mut account for only a subset of the phage resistance mechanisms underlying phage resistance, and genomic regions subject to deletion are also variable [80,84,85]. Consequently, a primary area of focus for future research will involve devising strategies for obtaining the desired form of resistance with increased frequency and broader generality. This objective may be achieved through the co-evolution of phages and bacteria to improve phage infectivity [94,95], or by developing engineered phages utilizing synthetic biology platforms [96,97]. Employing such strategies would allow us to develop phage cocktails that are designed to counter the multiple mechanisms of phage resistance, and to guide the acquisition of phage resistance along a specific direction.

To date, most clinical trials and case reports have not included evaluations of phage susceptibility pre- and post-treatment. Furthermore, apart from the ΦOMKO1 case [90], no clinical trials have been designed specifically to induce a fitness trade-off, or to utilize antibiotics based on the molecular mechanisms underlying trade-offs between phages and antibiotics. It would therefore be desirable to have a series of rigorously designed clinical trials in the future. These trials will not only serve to confirm the clinical applicability of molecular insights obtained in laboratory studies, but also to comprehensively validate the viability of this concept. We anticipate the following three primary strategies for the clinical application of phages to the enhancement of therapeutic outcomes in the future (Figure 4): (1) Utilizing phage cocktails for treatment to minimize the occurrence of phage-resistant variants. It is important that the phages in the cocktail recognize distinct receptors. New phage cocktails will be individually tailored to address emerging phage-resistant variants: (2) Initially employing a phage (or phage cocktail) in combination with antibiotics that exhibit a trade-off relationship with the phage (or phage cocktail). (3) Starting treatment with the phage (or phage cocktail) alone, with the goal of targeting desired phage resistance. This approach allows clinicians to preserve the use of antibiotics. If phage-resistant variants develop, treatment would then be switched to use of antibiotics with a trade-off relationship with the phages. To effectively implement such phage therapy, we need to develop a diagnostic system capable of rapidly confirming the genetic profile of target clinical isolates. This system will aid in designing phage cocktails with a high inhibitory effect (by recognizing distinct receptors) and determining which antibiotics have a trade-off relationship with which specific phages, enabling their effective use. In addition, phage-derived endolysins and depolymerases show great potential in enhancing phage efficiency, aiding in the removal of biofilms and persisters, which can limit the maximum effects of phage therapy [18,98]. Combining these with phages also could accelerate the development of phage-based therapy.

## 8. Conclusions

Phages have once again come into focus as a powerful approach for combating antimicrobial-resistant bacteria. Additionally, as the fitness trade-offs between phages and antibiotics are becoming clearer, phages have been identified as “old and new” approaches to resensitizing bacteria to antibiotics. From this perspective, phages are not solely antimicrobial agents for treating bacterial infections, but also powerful agents that could extend the lifespan of classical antibiotics and contribute to effective infection control of antimicrobial-resistant bacteria. It will be essential to clarify the molecular basis of these fitness trade-offs and design appropriate clinical trials based on their applicability. This type of phage-based therapy is expected to develop as a promising approach in combination with antibiotics, providing robust options in this era of phage “re”-discovery in the domain of modern medicine.

## Figures and Tables

**Figure 1 ijms-24-15628-f001:**
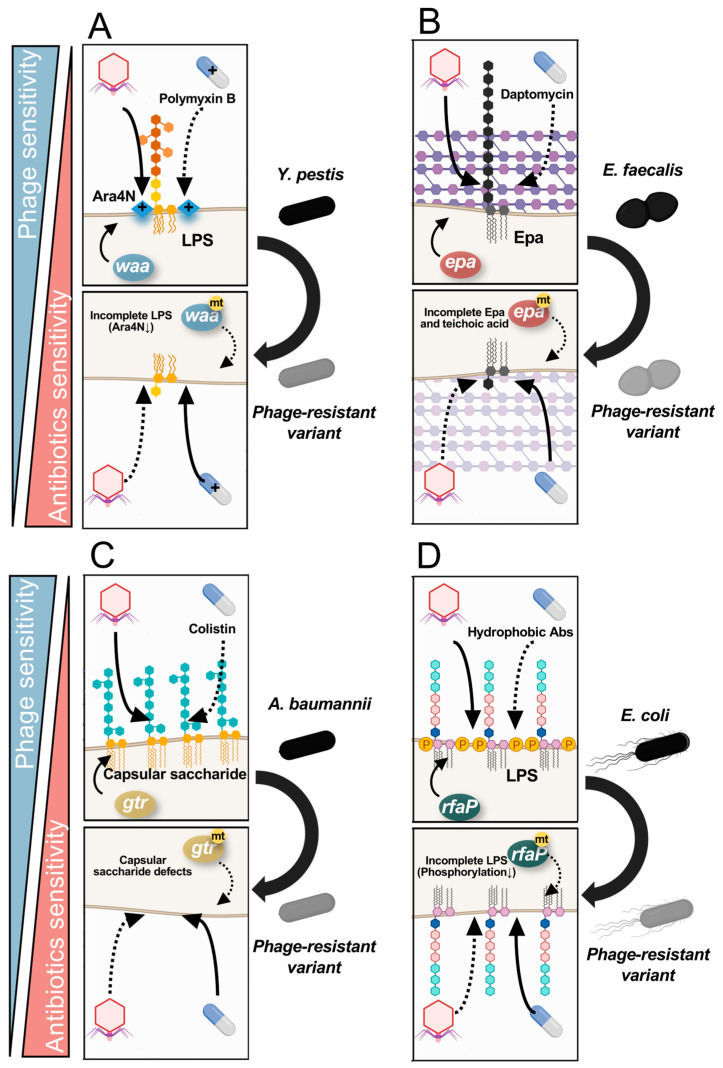
**Molecular mechanisms for sensitization to antibiotics via saccharide-associated phage resistance.** (**A**) A trade-off between polymyxin B sensitivity and phage sensitivity, which has been attributed to mutations in *waa*, a set of genes that regulate the length of the chain of LPS, has been reported in *Y. pestis*. Notably, it is estimated that an incomplete LPS core structure results in loss of phage susceptibility, and the LPS content of positively charged Ara4N is also reduced, resulting in increased susceptibility to positively charged polymyxin B. (**B**) A trade-off between daptomycin and phage susceptibility, which has been attributed to mutations in *epa*, a set of genes that biosynthesize Epa, has been reported in *E. faecalis*. *epa* has been suggested to contribute to the synthesis of teichoic acid in addition to Epa biosynthesis. The trade-off between phage resistance and daptomycin resistance is considered to be due to defects in the structure of teichoic acid and Epa. (**C**) A trade-off between colistin and phage susceptibility, which has been attributed to mutations in genes such as *gtr* and *gpi*, genes that are responsible for the synthesis of capsular polysaccharides, has been reported in *A. baumannii*. Deletion of the capsular polysaccharide causes the phage to lose its receptor, but the bacterium loses its barrier to colistin access, which is estimated to increase its susceptibility to colistin. (**D**) A trade-off between hydrophobic antibiotics and phage susceptibility, which has been attributed to mutations in the *rfpP* gene that is responsible for phosphorylation of LPS, has been reported in *E. coli*. *rfaP* mutations are considered to reduce phosphorylation and improve access and increase susceptibility to hydrophobic antibiotics. mt: mutation; Abs: antibiotics.

**Figure 2 ijms-24-15628-f002:**
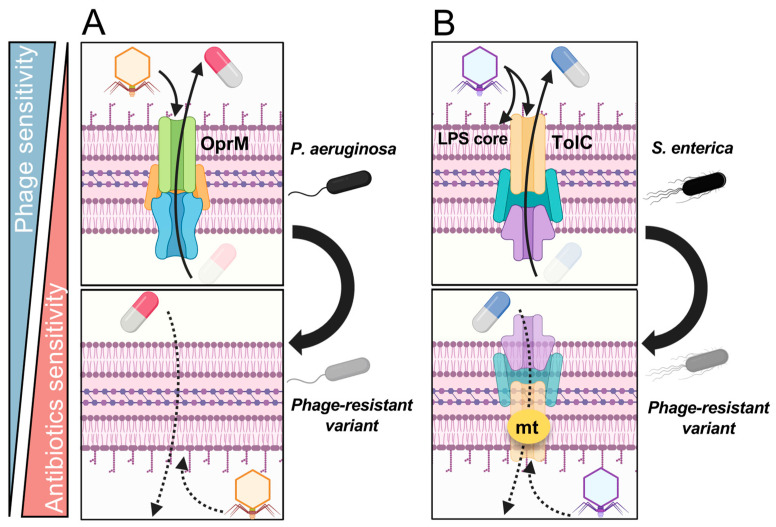
**Molecular mechanisms for sensitization to antibiotics via drug efflux transporter-associated phage resistance.** (**A**) In *P. aeruginosa*, a phage targeting OprM is known to cause functional loss of the drug efflux pump along with phage resistance. When such phage resistance occurs, sensitivity to antibiotics, which are substrates for the drug efflux pumps composed of OprM (i.e., MexAB-OprM and MexXY-OprM) is restored. (**B**) In *S. enterica*, a phage targeting TolC (which simultaneously recognizes the LPS core) is known to cause functional loss of the drug efflux pump along with phage resistance. When such phage resistance occurs, sensitivity to antibiotics, which are substrates for the drug efflux pumps composed of TolC, is restored. mt: mutation.

**Figure 3 ijms-24-15628-f003:**
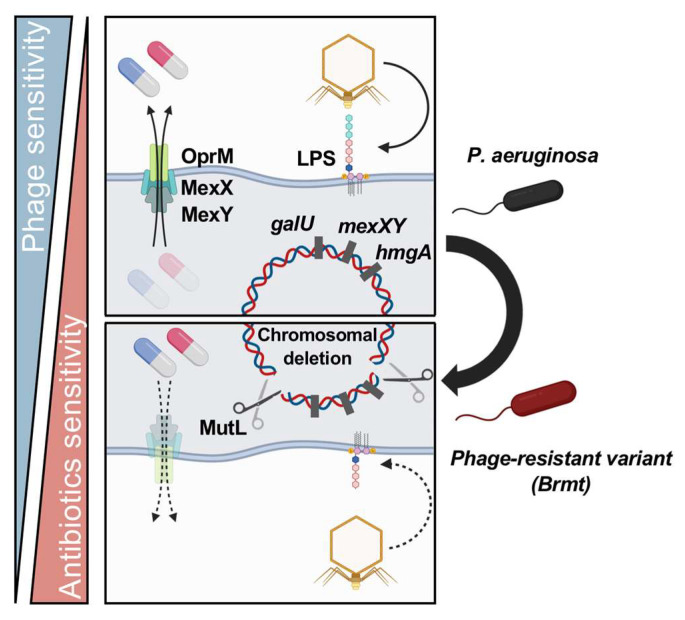
**Molecular mechanisms for sensitization to antibiotics via large chromosomal deletions-associated phage resistance.** Large-scale deletions of chromosomes by molecules such as MutL can result in phage resistance. In such cases, the deleted regions contain genes encoding phage receptors. Notably in *P. aeruginosa*, *galU*, which is important for complete LPS biosynthesis, and *hmgA*, a homogentisic acid-metabolizing enzyme, are deleted simultaneously in a large chromosomal deletion, resulting in phage-resistant variants with brown pigmentation (brmt). It has also been reported that the mexX and mexY genes encoding drug efflux transporters are located between *galU* and *hmgA*; thus, it has been observed that when phage resistance is acquired by chromosomal deletion in a *galU*-dependent manner, sensitivity to quinolones, the substrates of MexXY-OprM, is restored.

**Figure 4 ijms-24-15628-f004:**
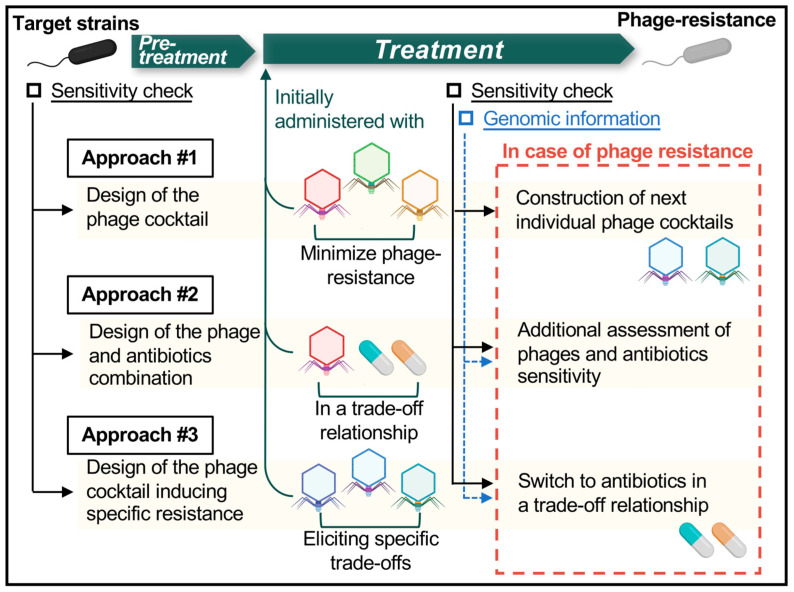
**Three major approaches to the future clinical application of phages to control phage resistance.** In order to control phage-resistant variants, it is necessary to reduce their frequency of occurrence. It is therefore desirable to prepare a phage cocktail that recognizes distinct receptors and that minimizes the incidence of phage resistance (Approach 1). However, as shown in Table 2, phage-resistant variants might still occur, and in such cases, a new phage cocktail for phage-resistant variants must be individually prepared. On the other hand, as our understanding of the molecular basis of the phage and antibiotic susceptibility trade-off increases, it will be possible to optimize phage-antibiotic combinations that maximize the beneficial effects of the trade-off relationship and to administer this combination from the beginning of phage therapy (Approach 2). For a strategy that preserves antibiotics, designing and administering a phage cocktail with the intention of a trade-off may lead to successful control of infections by inducing the intended specific trade-off and switching from phage to antibiotics to which the bacteria have restored susceptibility, even if phage-resistant variants emerge (Approach 3). In any case, it is desirable to develop a platform that can effectively provide the characteristics and genomic evidence of phage-resistant variants during phage therapy. Using such a platform will increase the efficacy of phage and antibiotic selection.

**Table 1 ijms-24-15628-t001:** Human clinical trials of phage therapy for bacterial infections.

Target Bacteria	Disease	Phage and Dose	Trial Design and Treatment Method	Pre-Phage Sensitivity	Post-Phage Sensitivity	Phage Resistance Analysis	Outcome	Ref.
*P. aeruginosa*	Otitis	6 phages (10^9^ PFUs) Single dose —intra-aural administration	Treatment group —12 individuals received phages Placebo group —12 individuals received glycerol-PBS buffer Double blind	Checked	Not checked	Not performed	Significant differences in terms of clinical improvement scores and levels of *P. aeruginosa* were observed between the phage-treated group and the placebo control. No adverse effects.	[42]
*E. coli*	Diarrhea	11 T4-like phages (3.6 × 10^8^ PFUs) or ColiProteus (1.4 × 10^9^) Three times/day for 4 days (12 doses) —oral administration without gastric neutralization	Treatment group —39 individuals received 11 T4-like phages —40 individuals received ColiProteus Placebo group —41 individuals received rehydration solution Double blind	Not checked	Checked	Not performed	There were no indications of phage efficacy on diarrhea parameters in the treatment group, and as a result, the trial was terminated early. No adverse effects.	[43]
*P. aeruginosa*	Burn wound infection	12 phages (2 × 10^7^ PFUs were expected but 200–2000 PFUs were observed) Once/day for 7 days (7 doses) —topical administration	Treatment group —12 individuals received phages Placebo group —13 individuals received standard of care (1% sulfadiazine silver) Blind	Not checked	Checked	Not performed	Phage treatment resulted in a slower reduction in bacterial load in burn wounds compared to the standard of care. The administered phage dose was significantly lower than the expected dose, leading to the trial being halted.	[44]

**Table 2 ijms-24-15628-t002:** Clinical cases in which phage therapy failed or treatment strategy had to be changed due to occurrence of phage-resistant variants.

Case	Year	Institution and Country	Target Bacteria	Disease	Patient	Phages	Dose and Treatment	Occurrence of Phage Resistance	Impact of Phage Resistance	Ref.
1	2016	University of California, San Diego, USA	*A. baumannii*	Necrotizing pancreatitis	Male, 68-years-old	Phage cocktail (ΦPC, 4 phages, Therapeutic dose was not applicable -used for intracavitary wash) Phage cocktail (ΦIV, 4 phages, 5 × 10^9^ PFUs)	ΦPC: percutaneous catheters drained the pseudocyst cavity, the biliary cavity, and a third intra-abdominal cavity for 18 weeks ΦIV: intravenous administration for 16 weeks	TP3, which was isolated 8 days after the initial administration of phage cocktails (ΦPC and ΦIV), exhibited resistance to both of the cocktails	Alternative personalized phage cocktails were constructed against resistant variants.	[38,39]
2	2016	Eliava Phage Therapy Center, Georgia	*S. aureus*	Netherton syndrome	Male, 16-years-old	Staphylococcus phage Sb1 and Pyo phage cocktail contains phages active against *S. aureus*	Oral administration of both 10 mL once a day and spray (Pyo) and cream (Sb1) for 40 days in total, next second 3 month period, phages for 2 weeks, with 2 week breaks in between treatment courses	After 3 months of treatment, phage sensitivity profiles showed resistance to Pyo phage cocktail	Alternative personalized phage cocktails were constructed against resistant variants.	[45]
3	2017	Eliava Phage Therapy Center, Georgia	*P. aeruginosa*	Cystic fibrosis	Male, 43-years-old	Pyo and Intesti phage cocktails (both of which contain phages active against *P. aeruginosa*)	8 mL taken orally once a day and 2 mL via nebulizer	Throughout the initial trimester of treatment for 3 months, the patient’s *P. aeruginosa* strains gradually developed resistance to Pyo and Intesti phage cocktails	Alternative personalized phage cocktails were constructed against resistant variants.	[46]
4	2019	Shanghai Public Health Clinical Center, China	*K. pneumoniae*	Urinary tract infection	Female, 63-years-old	Phage cocktail I (5 phages, 5 × 10^8^ PFUs/mL of each phage) Phage cocktail II (5 phages, active against phage-resistant variants against cocktail I)	Phage cocktail I: 10 mL bladder irrigation, daily for 5 days Phage cocktail II: 50 mL bladder irrigation, daily for 5 days	In two rounds of phage therapy (1st using phage cocktail I, 2nd using phage cocktail II), phage-resistant mutants developed within days in both of rounds	As reemergence of phage-resistant was repeated, novel therapeutic strategies are needed. Eventually, by combination of phage and antibiotics, the patient was cured.	[47]
5	Reported in 2020	University of California, San Diego, USA	*P. aeruginosa*	Lower respiratory tract infections	Male, 67-years-old	Phage cocktail AB-PA01 (4 phages, 4 × 10^9^ PFUs/mL)	Intravenous administration every 6 h and nebulized every 12 h for 4 weeks.	One isolate resistant to AB-PA01 appeared after cessation of AB-PA01	A personalized approach was required to identify new phages active against the resistant isolates.	[39,48]
6	Reported in 2020	University of California, San Diego, USA	*P. aeruginosa*	Ventricular assist device infection	Male, 82-years-old	Phage cocktail SDSU1 and 2 (2 phages, respectively, 2 × 10^5^ PFUs/mL),	Intravenous administration every 8 h for 6 weeks, +1-time intraoperative dose followed by one phage alone for 10 days and followed by SDSU2 IV every 12 h for 3 weeks	During phage therapy, phage resistance developed.	Additional new phages, in a personalized treatment approach, were required to account for resistant isolates.	[39]
7	Reported in 2021	Johns Hopkins University, USA	*M. abscessus*	Refractory abscessus lung disease	N/A, 81-years-old	Phage cocktail (3 phages, 2 × 10^9^ PFUs)	Intravenous administration twice daily for 6 month	Post-treatment *M. abscessus* isolates demonstrated emergence of phage resistance against one phage in the cocktail	Although phage resistance was observed, phage therapy failed mainly due to potent antibody-mediated neutralization.	[49]
8	2021	Not specified, Spain	*P. aeruginosa*	Prosthetic vascular graft infection	Male, N/A	Phage cocktail (3 phages, 10^7^ PFUs/mL)	Intravenous administration of 70 mL phage cocktail, once a day in a 6-h infusion for 1 week	The post-phage therapy isolate exhibited phage resistance	Failed to eradicate the infection by the phage cocktail with ceftazidime-avibactam. After phage therapy, a proximal vascular prosthesis replacement combined with antibiotic therapy was performed, and patient remains asymptomatic.	[50]

**Table 3 ijms-24-15628-t003:** Clinical phage therapy cases in which the combination of antibiotics was successful or inhibited the development of phage-resistant variants.

Case	Year	Institution and Country	Target Bacteria	Disease	Patient	Phages	Combined Antibiotics	Outcomes	Ref.
1	2015	Hannover Medical School, Germany	*S. aureus* *E. faecium* *P. aeruginosa*	Prosthetic infection after aortic arch replacement	Male, 52-years-old	Phage cocktail (CH1 against *S. aureus*, Enf1 against *E. faecium*, PA5 and PA10 against *P. aeruginosa*, 10^8^ PFUs/mL), topical administration via drainage and intraoperatively. One oral administration	Gentamicin and daptomycin (with drainage), cefepime, daptomycin, linezolid, tobramycin (iv)	*S. aureus*, *E. faecium* and *P. aeruginosa* were not detected after phage and antibiotic treatment.	[93]
2	2016	Yale University, USA	*P. aeruginosa*	Chronic infection in aortic Dacron graft with aorto-cutaneous fistula.	Male, 80-years-old	Phage OMKO1 (10^8^ PFUs), topical administration	Ceftazidime	Clinically successful. Phage revealed bactericidal activity while also eliciting the emergence of phage resistance, which correlated with a resensitization to the antibiotic.	[90]
3	2016	Hannover Medical School, Germany	*K. pneumoniae*	Lung infection during drug-induced immuno-suppression after heart transplantation	Male, 40-years-old	Phage cocktail (KPV811 and KPV15 against *K. pneumoniae*, 10^8^ PFUs/mL), Inhalation and topical administration via nasogastric tube	Ceftazidime, linezolid, avibactam, meropenem, cotrimoxazole, tobramycin (iv), colistin (inhalation)	*K. pneumoniae* was not detected in bronchial lavage after phage and antibiotic treatment.	[93]
4	2017	Hannover Medical School, Germany	*S. aureus*	Chronic vascular graft infection after aortic arch replacement. Replacement	Male, 59-years-old	Phage CH1 (10^9^ PFUs/mL), topical administration via drainage	Rifampicin, flucloxacillin (iv)	*S. aureus* was not detected after phage and antibiotic treatment.	[93]
5	2017	Hannover Medical School, Germany	*S. aureus*	Fulminant pleural empyema after left-ventricular assist device implantation	Male, 62-years-old	Phage CH1 (10^9^ PFUs/mL), topical administration via drainage	Daptomycin (iv)	Although *S. aureus* was not detected after phage and antibiotic treatment. The patient died due to transplant failure after 20 months of transplantation.	[93]
6	2017	Hannover Medical School, Germany	*S. aureus*	Chronic left-ventricular assist device infection	Male, 51-years-old	Phage cocktail (Sa30, CH1, SCH11 and SCH111, 10^9^ PFUs/mL). Topical administration via drainage, intranasal and oral administration	Daptomycin (iv)	The level of *S. aureus* was reduced to 10^−2^ in the drainage fluid and eradicated completely from nose and throat. Occurrence of phage-resistant variants was not observed throughout phage therapy. Nevertheless, the patient died 1.5 months after.	[93]
7	2018	Saint-Luc University Hospital, Belgium	*P. aeruginosa*	Sepsis after liver transplantation	Male, toddler	Phage cocktail, BFC1, containing one *S. aureus* phage (ISP) and two *P. aeruginosa* phages (PNM and 14-1), intravenous, intralesional and intra-abdominal administration	Vancomycin, gentamycin, colistin, aztreonam, cotrimoxazole, fluconazole	The occurrence of bacterial phage resistance did not result in therapeutic failure and in vitro phage-antibiotic synergies were clearly observed.	[88]
8	2018	Hannover Medical School, Germany	*E. coli*	Sternal wall healing disorder after mitral valve replacement and aortocoronary bypass	Female, 66-years-old	Phage cocktail (ECD7 and V18, 4 × 10^10^ PFUs/mL), intraoperatively administration mixed with fibrin glue	Clindamycin (Oral)	Phage mixture with fibrin glue allowed for the sustained release of phages to infected sites. The wound completely healed and *E. coli* was no longer detected after phage administration.	[93]
9	2018	Hannover Medical School, Germany	*P. aeruginosa*	Sternal wound abscesses after double lung transplantation	Male, 13-years old	Phage cocktail (PA5 and PA10, 4 × 10^10^ PFUs/mL), intraoperatively administration mixed with fibrin glue	Colistin, ceftazidime, avibactam (iv)	Phage mixture with fibrin glue allowed for the sustained release of phages to infected sites. The wound completely healed and *P. aeruginosa* was no longer detected after phage administration.	[93]
10	Reported in 2019	University of California, San Diego, USA	*P. aeruginosa*	Cystic fibrosis	Female, 26-years-old	Phage cocktail (AB-PA01, 4 phages), intravenous administration	Azithromycin, ciprofloxacin, doripenem, piperacillin-tazobactam, vancomycin	Clinical success. Although a phage-resistant isolate temporarily emerged, it did not pose a therapeutic problem. This is likely an effect of the combination of antimicrobial agents.	[39,89]
11	Reported in 2019	Hadassah-Hebrew University Medical Center, Israel	*A. baumannii* *K. pneumoniae*	Trauma-related left tibial infection	Male, 42-years-old	Phage cocktail (AbKT21phi3 against *A. baumannii* and KpKT21phi1 against *K. pneumoniae*), intravenous administration	Meropenem, colistin	Clinically successful and the patient’s leg did not have to be amputated. No phage-resistant variants emerged in the patient, likely because the combination of antibiotics and phages was very effective.	[91]

## Data Availability

No new data were created or analyzed in this study. Data sharing is not applicable to this article.

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
