# Peer review of "Fitness Trade-Offs between Phage and Antibiotic Sensitivity in Phage-Resistant Variants: Molecular Action and Insights into Clinical Applications for Phage Therapy"

_ijms, 2023, doi:10.3390/ijms242115628_

Round 1
Reviewer 1 Report
In this manuscript, the authors have broadly discussed the success and failure of bacteriophage therapy, primarily focusing on the emergence of phage-resistant bacteria. They also addressed the trade-off between phage-resistant bacteria and antibiotic susceptibility. Towards the end, they delved into future strategies and directions for phage-based therapy, considering the aspect of fitness trade-offs. Overall, this is a well-written review. The table and pictorial representations of content are highly informative and facilitate a quick understanding of the topic.
I have one minor comment regarding the “virology of phages” section, second paragraph's first line: not all bacteriophages possess a head and tail structure, such as those without a tail and filamentous phages
Author Response
Manuscript ID ijms-2675980 entitled “Fitness Trade-offs between Phage and Antibiotic Sensitivity in Phage-resistant Variants: Molecular Action and Insights into Clinical Applications for Phage Therapy” by J Fujiki, et al.
Responses to the comments from the reviewers:
Reviewer 1
Comments:
In this manuscript, the authors have broadly discussed the success and failure of bacteriophage therapy, primarily focusing on the emergence of phage-resistant bacteria. They also addressed the trade-off between phage-resistant bacteria and antibiotic susceptibility. Towards the end, they delved into future strategies and directions for phage-based therapy, considering the aspect of fitness trade-offs. Overall, this is a well-written review. The table and pictorial representations of content are highly informative and facilitate a quick understanding of the topic. I have one minor comment regarding the “virology of phages” section, second paragraph's first line: not all bacteriophages possess a head and tail structure, such as those without a tail and filamentous phages.
Response:
We greatly appreciate the reviewer’s valuable comments on our manuscript. As the reviewer pointed out, we improved our manuscript in the “virology of phages” section, second paragraph's first line as described below: All phages package their genetic material within a capsid, and a significant number of phages possess a tail which is essential for attachment to host cells and for injection of their genome into the host.
Reviewer 2 Report
The manuscript seems to be well-structured and covers both clinical applications and future perspectives. The integration of phage therapy with antibiotics and the exploration of their synergistic effects is of potential significance in the medical field. The topic appears to be timely and relevant, especially considering the rise in antimicrobial-resistant bacteria. This manuscript provides a comprehensive overview of phage therapy within the context of antibiotic resistance and presents key concepts and developments in the field.
Please check these minor suggestions below:
Please make sure the citation style was maintained for consistency.
Line 66: Consider breaking this sentence for clarity. "The ongoing evolutionary competition between bacteria and phages has driven the development of diverse biological systems. This has contributed to the remarkable diversity observed in these microorganisms."
Line 99: "structures exposed on bacterial surfaces" -> "structures on bacterial surfaces"
Line 106: Change "living, non-visible viruses" to "invisible living entities"
Line 108-109: The information here feels redundant considering similar data was given in the introduction. Consider revising for brevity.
Line 119-120: consider changing the term or add phrase ‘during the Cold War era’ for better understanding for readers, for example… "where access to antibiotics was limited by the "Iron Curtain”," -> "where access to antibiotics was restricted during the Cold War era,"
Line 125-126: "because phages have distinct bactericidal mechanisms" -> "given their unique bactericidal mechanisms"
Line 129-134: Consider shortening this example for brevity. Although it provides an insightful clinical scenario, it can be condensed.
Line 139: "phage medicines through magistral preparation" -> "phage treatments using magistral preparations"
Line 144-145: "primarily because of the absence of pre-phage sensitivity testing" -> "largely due to the lack of prior phage sensitivity tests"
Line 174-177: For better clarity and to avoid redundancy: "In two separate cases in China and Spain (Cases 4 and 8; Table 2), the emergence of phage-resistant bacteria mandated exploring alternative therapeutic strategies, which may or may not have incorporated phages (50, 51)."
Some sentences appear somewhat redundant, though this does not significantly impact comprehension. I recommend that the authors review the manuscript for redundancy and ensure proper citation formatting.
Overall, the manuscript is well-written.
Reviewer 3 Report
The review article "Fitness Trade-offs between Phage and Antibiotic Sensitivity in Phage-resistant Variants: Molecular Action and Insights into Clinical Applications for Phage Therapy " provides recent developments in the field of phage-antibiotics combined treatment options for effective treatment of antibiotic resistant strains. The review is well written and provides updated references, but lacks few recent developments in this field!
- Firstly there is not much reference about puage bound enzymes like endolysins or depolymerases which are important for effective phage therapy!
https://www.mdpi.com/2079-6382/10/2/124
- Secondly, recent work in the field of phage in-vivo therapy has shown that phage bound depolymerases are effective against biofilm removal irrespective of its specificity! Such approach of phage as a enzyme bound nanoparticle must also be discussed!
https://www.mdpi.com/2073-4409/12/3/344
The English language is of better quality and only need minor spell check.
